# mTOR Contributes to the Proteome Diversity through Transcriptome-Wide Alternative Splicing

**DOI:** 10.3390/ijms232012416

**Published:** 2022-10-17

**Authors:** Sze Cheng, Naima Ahmed Fahmi, Meeyeon Park, Jiao Sun, Kaitlyn Thao, Hsin-Sung Yeh, Wei Zhang, Jeongsik Yong

**Affiliations:** 1Department of Biochemistry, Molecular Biology and Biophysics, University of Minnesota Twin Cities, Minneapolis, MN 55445, USA; 2Department of Computer Science, University of Central Florida, Orlando, FL 32816, USA

**Keywords:** mTOR signaling, alternative splicing, post-transcriptional gene regulation, functional proteome

## Abstract

The mammalian target of rapamycin (mTOR) pathway is crucial in energy metabolism and cell proliferation. Previously, we reported transcriptome-wide 3′-untranslated region (UTR) shortening by alternative polyadenylation upon mTOR activation and its impact on the proteome. Here, we further interrogated the mTOR-activated transcriptome and found that hyperactivation of mTOR promotes transcriptome-wide exon skipping/exclusion, producing short isoform transcripts from genes. This widespread exon skipping confers multifarious regulations in the mTOR-controlled functional proteomics: AS in coding regions widely affects the protein length and functional domains. They also alter the half-life of proteins and affect the regulatory post-translational modifications. Among the RNA processing factors differentially regulated by mTOR signaling, we found that SRSF3 mechanistically facilitates exon skipping in the mTOR-activated transcriptome. This study reveals a role of mTOR in AS regulation and demonstrates that widespread AS is a multifaceted modulator of the mTOR-regulated functional proteome.

## 1. Introduction

mRNA splicing is a critical co-transcriptional process to produce uninterrupted coding DNA sequences (CDS) for protein translation [1]. Alternative splicing (AS) can occur by alternating the inclusion of exons or part of exons to produce mature mRNAs [2,3,4]. Thus, AS diversifies the proteome by potentiating the production of multiple protein isoforms from a gene [5,6]. Although AS serves as an essential layer of post-transcriptional regulation of gene expression in eukaryotes [7,8], much remains to be explored concerning its regulation and impact on the resulting proteome. 

Mammalian target of rapamycin (mTOR) is a serine/threonine kinase and is the major component of the mTOR Complex 1 (mTORC1) and 2 (mTORC2) [9,10,11]. The heterodimeric tuberous sclerosis complex composed of TSC1 and TSC2 is a negative regulator of mTORC1. mTOR promotes the translation of a subset of mRNAs that contain the 5′-terminal oligopyrimidine tract (5′-TOP) [12,13,14,15]. mTOR was also shown to regulate select RNA-binding proteins (RBPs) and functions in post-transcriptional regulation [16,17,18]. Hyperactivation of mTOR leads to transcriptome-wide alternative polyadenylation (APA) in the 3′-untranslated regions (3′-UTR) of transcripts and affects diverse cellular pathways [19,20]. mTOR activation also prefers the expression of splicing factor U2AF1a isoform to that of the b isoform and renders U2AF1a-dependent AS events [21]. 

In this study, we investigated mTOR-driven transcriptome changes in alternative splicing and their impact on the resulting proteome using various cellular models. We found that widespread exon skipping by mTOR activation diversifies the proteome by changing functional domains, post-translational modifications, and protein stabilities.

## 2. Results

### 2.1. mTOR Activation Leads to Exon Skipping

To better understand the transcriptomic features driven by mTOR signaling, we profiled the transcriptome of various mammalian cells with low and high mTOR contents: *Tsc1-/-* mouse embryonic fibroblasts (MEFs), human breast cancer cell line MCF7, and MDA-MB361 treated with DMSO (mock) or Torin 1, a potent inhibitor of mTOR. From our initial interrogation focused on the RNA processing pathway, we found numerous RNA-binding proteins (RBPs) involved in alternative splicing (AS) and alternative polyadenylation to be differentially expressed (Figure 1A). The heat map showed that a cluster of RBPs is consistently upregulated among all cells with high mTOR activity. Based on a literature search, many of these RBPs (bolded in the enlarged heatmap) have demonstrated either overexpression in cancer or play tumor-promoting roles [22,23,24,25,26,27,28,29,30,31,32,33,34,35,36]. These differentially expressed RBPs modulated by mTOR signaling suggest a comprehensive role of mTOR in the transcriptome by post-transcriptional regulatory pathways that could be important for cancer progression. We then profiled the differential AS events using the custom-developed AS-Quant pipeline [37]. Strikingly, we found that chemical inhibition of mTOR drives exon skipping in both MEFs and human breast cancer cell lines. The exon skipping occurrence is most evident in the cassette exon category: among the affected AS events, exon inclusion in cassette type AS was preferred (452 events; 96%) in Torin 1-treated *Tsc1-/-* MEFs (Figure 1B). This trend of exon inclusion upon mTOR inhibition was consistent with other types of AS events with varying degrees (59% in alternative 5′ SS site and 82% in alternative 3′ SS site) (Figure 1B). We noticed that the number of exon inclusions varied among the profiled cellular models (60% in breast cancer transcriptome compared to 90% in *Tsc1-/-* MEF transcriptome). This discrepancy may come from the degree of mTOR activation in these cell lines examined: *Tsc1-/-* MEFs are genetically programmed to hyperactivate mTOR signaling while MCF7 and MDA-MB361 cells have a reduced level of mTOR hyperactivation (Appendix A). Thus, upon mTOR inhibition, changes in AS events are maximized in the *Tsc1-/-* system compared to breast cancer cells (Appendix A). In a similar context, when the same *Tsc1-/-* transcriptome was analyzed against the WT (basal mTOR activity) transcriptome, it showed fewer exon inclusion-biased AS events compared to that of the *Tsc1-/-* transcriptome with Torin 1 treatment. We further delved into the nucleotide contexts surrounding the 3′-splice site and found that mTOR activation promotes the skipping of the cassette exon that contains a strong, longer stretch of the poly-pyrimidine tract (−3 to −20 position) and a weak poly-pyrimidine tract with biased C at the −1 position in the alternative 3′-splice site in both *Tsc1-/-* MEFs and human cancer cells (Figure 1C). Notably, in the case of alternative 3′-splice site selection by mTOR activation, the consensus nucleotide context of the skipped 3′-splice site showed a strong and longer poly-pyrimidine tract similar to that of skipped cassette exon, suggesting that a suppressive role of poly-pyrimidine tract in the 3′-splice site selection is widespread in the mTOR-activated transcriptome. From the transcriptome-wide AS splicing by mTOR activation, we determined the overlap in AS genes with genes that showed differential gene expression (Figure 2A). We found that the majority of AS genes did not overlap with the differential expressed genes, suggesting that gene regulation by mTOR signaling occurs at many levels: AS and differential gene expression. Additionally, we performed enrichment analysis of the mTOR-mediated AS genes and found that they are involved in pathways, such as the spliceosome, ubiquitin-mediated proteasome degradation, and many other important biological processes (Figure 2B). Compared to the list of enriched pathways from the mTOR-mediated AS gene list, the mTOR-mediated DGE gene list featured a largely distinct list of pathways. Interestingly, four pathways were found to be overlapped between the AS and DGE gene list: the p53 signaling pathway, small cell lung cancer, oocyte meiosis, and viral carcinogenesis. However, the individual genes impacted by AS and DGE were overwhelmingly different in those overlapping pathways, suggesting that AS and DGE regulated by mTOR serve as independent paths to impact the same biological processes. Further examinations of affected AS events showed that the majority of them occur in the CDS of mRNA (65–68%) while 26–27% of them are found in the 5′-UTR (Figure 2C), suggesting that the majority of mTOR-mediated AS events impact the protein coding region and can have a functional impact in the proteome. Next, we validated mTOR-regulated AS events on selected genes using PCR and analytical gel electrophoresis. Consistently, all tested genes showed exon skipping upon mTOR activation (*Tsc1-/-* MEFs mock vs. Torin 1 treatment) (Figure 2D). These findings suggest that mTOR activation escalates the frequency of exon skipping during pre-mRNA splicing leading to preferential expression of exon-skipped isoforms in the transcriptome.

### 2.2. Increased Intron Retention of Ribosomal Genes by mTOR Activation 

Out of all the categories of AS, intron retention poses the most detriment toward altering the protein structure. Notably, the AS-Quant pipeline did not detect many differentially expressed, annotated intron-retained transcripts. It has been reported by several studies that global intron retention serves as a transcriptional control during granulocyte differentiation and T-cell activation [39,40]. Thus, we strategized a computational pipeline (Figure 3A) to identify novel de novo intron-retained transcripts. With our intron-retention pipeline, we detected a significant amount of differentially expressed previously unannotated intron-retained transcripts upon mTOR inhibition (Figure 3B). Consistent with these findings, a comparison of other pairs of AS using the datasets from WT, *Tsc1-/-* MEFs, and human breast cancer cell lines MCF7 and MDA-MB361 showed an overall differential intron retention upon mTOR inhibition. (Figure 3B). Selected differentially intron-retained transcripts were validated and semi-quantitated by RT-PCR gel electrophoresis (Figure 3C). When intron retention occurs in the coding region of the mRNA, there are three possible types of translational outcome: (1) protein truncation; (2) in-frame exonization of the intron; and (3) out-of-frame exonization of the intron. In our datasets, we found that the majority of these intron retention events regulated by mTOR yielded the production of truncated proteins as a stop codon is frequently introduced in the retained intron (Figure 3D). While a small number of cases did not contain an intronic stop codon, the inclusion of the intron in those cases generated a frameshift in translation, resulting in the generation of a protein isoform with a new C-terminal tail. Functionally, genes with intron-retained transcripts with preferential expression in high mTOR contents are largely enriched in the ribosome and translation pathways (Figure 3D), which suggests that intron retention in ribosomal genes can serve as an additional regulatory pathway for ribosome biogenesis by mTOR signaling.

### 2.3. AS Contributes to the mTOR-Programmed Functional Proteome at Multiple Levels

In our dataset, we found frequent cases of AS located in 5′UTR or the coding region in which the skipping or the inclusion of the AS exon affects the translation initiation site, producing N-terminal truncated protein isoforms. We found that mTOR-promoted AS of *Sirt2*, *Mdm2*, and *Adnp* causes a frameshift in the original ORF and enforces the use of the next available translation initiation codon. N-terminal protein sequences are known to affect the protein degradation rate by providing signals for ubiquitination and proteolysis, known as the N-end rule [41]. To understand the functional relevance of mTOR-regulated N-terminally truncated AS isoforms, we investigated the differential protein stability caused by AS-induced methionine availability for translation initiation. To do this, we performed cycloheximide chase assays on Flag-tagged SIRT2, -MDM2, and –ANDP AS isoforms and found that the N-terminal truncated short isoform of these proteins is generally less stable than the long isoform, suggesting that AS coding the N-terminus sequence affects the stability of the protein isoforms (Figure 4A and Appendix A). Besides the N-terminal changes induced by AS, AS within the coding region can contribute to the gain or loss of peptide-sequence-encoding protein domains, which are important determinants for the function of the protein. Thus, we determined the proportion of AS exons located in the coding region that contributed to Pfam domain generations or disruptions. From this, we found that one-third of the AS exons in the CDS cause gain of loss of functional Pfam domains (Figure 4B) and 114 of these affected domains are associated with various GO terms (Figure 4C). As exon skipping generally removes coding sequences within the AS exon and results in a shorter isoform, we found that the majority of the domains affected by exon skipping disrupted the functional domain as peptides within the domains were lost. This finding suggests that the function of these exon-skipped protein isoforms may be different than their full-length counterpart as domain disruption could result in either of loss of function or gain of function in the proteome. Interestingly, AS cases encoding the Pkinase domain and Pkinase_Tyr domain were found to be gained in certain kinases and lost in other kinases within the mTOR-activated proteome, suggesting that mTOR can fine-tune different kinase function by modulating AS in different kinase genes. Additionally, from the few mTOR-preferred AS exon-inclusion cases, we identified domains for tRNA-synt-2b, Aph-1, and GAS2 that were generated or gained. The aminoacyl tRNA synthetase domain is known to be critical for its function in catalyzing the aminoacylation reaction for protein synthesis [42]. The upregulation of the exon-included aminoacyl tRNA synthetase with a functional Pfam domain may be a way by which mTOR ramps up the protein synthesis pathway. In addition to the change in Pfam domains, AS can also induce the generation or loss of amino acid residues that become post-translationally modified (PTM). mTOR activation has previously been shown to exhibit changes in the phosphoproteome, suggesting that mTOR can regulate proteome-wide PTMs [43,44]. We next asked whether mTOR-driven AS in CDS also affects proteome-wide PTMs [45,46,47,48]. We cataloged the unique peptide sequences belonging to each AS isoform and compared them to peptide sequences in published PTM proteomics [43,49,50,51,52,53,54]. In this proteome-wide search, we found that exon skipping could abolish existing sites for four interrogated PTMs in select genes and could also create new PTM sites in other genes, demonstrating that mTOR-modulated AS serves as a molecular scaffold for PTMs in functional proteomics (Figure 4D). Following the identification of these PTM site changes by mTOR-AS, we wanted to see if these sites have reported known functions from the literature. To highlight, one of the cases we found, the long SIRT2 isoform (NM_022432) creates a unique phosphoserine site that allows for cell-cycle-dependent chromatin localization while the short isoform (NM_001122765) missing the phosphoserine site cannot (Figure 4E). [55,56]. Another notable example is the long ARF2 isoform (NM_001025093) generated by exon 5 inclusion that contains a unique phosphoserine site crucial for the transcriptional activity of ARF2 [57]. Together, these data suggest that mTOR-induced exon skipping poses different aspects of functional relevance in the proteome through protein stability, Pfam domain changes, and PTM changes.

### 2.4. SRSF3 Promotes Exon Skipping in the mTOR-Activated Transcriptome

Previously, we reported the upregulation of splicing factor SRSF3 by mTOR-driven 3′-UTR APA and suggested a suppressive role of SRSF3 in mutually exclusive AS events [19]. To more broadly understand the effect of mTOR-activated SRSF3 upregulation on transcriptome-wide AS, we knocked down *Srsf3* expression in *Tsc1-/-* MEFs (Appendix A) and profiled AS events using RNA-Seq experiments (Figure 5A). Interestingly, we found that 57% (197/346) of cassette exon-inclusion cases and 43% (149/346) of cassette exon-skipping cases are preferred in the *Srsf3* knockdown cells, suggesting that SRSF3 has a slight bias towards promoting exon-skipping but can promote exon-inclusion to a sizable fraction. Further analysis of overlapping AS events by SRSF3 and mTOR activation in *Tsc1-/-* MEFs showed that there is a 14.2% (64/452 cases) overlap in exon skipping cases and a 19% (4/21) overlap in exon-inclusion cases (Figure 5B), suggesting that SRSF3 upregulation is responsible for mTOR-dependent exon skipping and inclusion. A similar observation was made when the *Srsf3* knockdown data were compared with the dataset of WT versus *Tsc1-/-* MEFs. In total, 13.8% of exon-skipping overlap and 11.6% of exon-inclusion overlap between the two datasets was observed (Appendix A). This suppressive role of SRSF3 on select genes was validated using RT-PCR (Appendix A). To gain an insight into the suppression mechanism, an in silico search was conducted to map the consensus binding sequence of SRSF3 across the suppressed exon regions. This approach identified two major patterns of SRSF3-binding: a cluster of SRSF3 consensus motifs located immediately upstream of the 3′-splice site and nearly the entire suppressed exon regions (Figure 5C). Together, our data suggest that mTOR upregulation of SRSF3 is partially responsible for the AS changes in the mTOR-activated transcriptome.

## 3. Discussion

The mTOR pathway is a central regulator of anabolic metabolism in cells and has been associated with many human disease pathogenic mechanisms [9]. mTOR was shown to regulate the translational regulation of TOP or TOP-like sequences containing mRNAs [59,60]. Recently, mTOR’s role in post-transcriptional gene regulation and proteome regulation has been expanded by its relevance to 3′-UTR APA [19]. Additionally, in our previous study, we found that mTOR-regulated isoforms with AS exons located in the 5′-UTR have differential translation efficiency, ultimately resulting in changes in the protein abundance [21]. Here, we further investigated the changes in transcriptomic features using RNA-Seq experiments and report that mTOR activation drives transcriptome-wide AS and modulates the proteome diversity. We found that mTOR-activated MEFs and human cancer cells generally produce more exon-skipping transcripts in the transcriptome, posing multifaceted functions in the proteome in the context of protein stability, functional domain changes, and post-translational modification site changes. These changes in the proteomics cumulated by exon-skipped protein isoforms can affect protein abundance and alter protein functions (Figure 5D). Thus, it is likely that mTOR plays a role in the regulation of gene expression from transcription to RNA processing to translation, making mTOR a major signaling pathway for the regulation of gene expression [61]. Interestingly, our data show that mTOR activation generally drives the skipping of exons or part of exons in splicing and prefers to produce a shorter transcript isoform expression from a gene. Suppressed cassette type exons or alternative 3′-SS showed the presence of a long polypyrimidine sequence stretch (approximately 15–20 nucleotides) in the upstream intron region of 3′-SS (Figure 1C). This distinct molecular signature around the 3′-SS suggests that mechanistically, various polypyrimidine sequence-binding proteins are activated or upregulated by mTOR and sterically interfere with the access of U2AF to the 3′-SS by binding to the long polypyrimidine tract. Supporting this idea, the upregulation of several poly-pyrimidine tract binding proteins, such as PTBP1, PTBP3, and HNRNPC was observed in the mTOR-activated transcriptome. In a similar context, it is noteworthy that SRSF3 knockdown promotes the inclusion of exons in our dataset (Figure 5A). Previous transcriptome-wide footprinting for SRSF3 showed that SRSF3 binds to exon regions and prefers penta-pyrimidine sequences [62]. Thus, mTOR-activation-dependent SRSF3 upregulation could suppress exon inclusion by competing for polypyrimidine tracts located upstream of the 3′-SS. It could also compete with splicing enhancers to bind exons and suppress exon inclusion in AS.

The expression of shorter transcript isoforms by AS can result in various output in the functional proteome. In our data analyses, one of the most striking features is that mTOR-driven AS renders the loss or gain of protein domains that could possess PTMs. Thus, mTOR signaling not only modulates PTM, particularly phosphorylation cascades in the proteome but also alters the profile of PTM sites in diverse proteins through AS. In the latter case, the profile of PTMs is not restricted to phosphorylation but extended to other types of modifications such as ubiquitination and acetylation, broadening the role of mTOR in the regulation of PTM-mediated protein functions beyond protein phosphorylation (Figure 4D). In the *Tsc1-/-* dataset, most AS events in the CDS did not cause the changes in Pfam domains (67%) while 33% of CDS AS events were linked to changes in the Pfam (Figure 4B). Although it is difficult to predict the functional outcomes of AS events without connections to Pfam alterations, an example of an *Sirt2* AS event indicates that such AS events with no Pfam changes would still affect the function of proteins through diverse regulatory mechanisms including PTMs. Among AS events that are associated with Pfams, it is interesting to observe that certain Pfams including PKinase and PKinase_Tyr are lost and gained across genes (Figure 4C). This suggests that in a global proteome perspective, a group of protein kinases and protein tyrosine kinases loses their activities while the other group of the same family of proteins gains their kinase activities. This shuffling of Pfam domains would reprogram these functional-domain-mediated protein functions in genes without affecting their gene expression levels. Interestingly, this Pfam-specific function shuffling occurs also in other Pfam domains such as Zip and 60KD_IMP, indicating that bioinformatics analysis of AS events and their associated Pfam changes would reveal a new perspective in the role of AS in functional proteomics (Figure 4C). Together, these results reveal that mTOR-driven AS not only modulates the functionality of individual genes but also selectively steers the function across the same family of proteins.

Compared to exon skipping that is more well-known, intron retention is less studied due to their common perception as background noise from RNA-seq expression analysis and poor documentation of such events in the current genome annotations [63,64]. Particularly, unannotated intron retention events cannot be analyzed using widely used AS pipelines such as AS-Quant [37] and DEXSeq [65]. In this study, we expanded the differential intron-retained transcript detection by mTOR using AS-Quant for de novo differential intron retention by mTOR. This add-on function to AS-Quant, which similarly functions like previous pipelines for the detection of unannotated intron retention, allowed us to uncover novel intron-retained transcripts regulated by mTOR signaling, and many of these upregulated intron-retained transcripts by mTOR activation were mRNAs for ribosomal proteins. Further investigation on the biological relevance of intron-retained transcripts for ribosomal proteins is warranted.

In summary, we investigated the global changes in alternative splicing by mTOR signaling and discovered transcriptome-wide AS changes and their functional relevance. Our study revealed a new function of mTOR in gene expression regulation and uncovered additional layers of proteome regulation via alternative splicing. Future studies will be needed to determine the pathological relevance of isoform switching by mTOR signaling in cancer and other diseases.

## 4. Materials and Methods

### 4.1. Cell Lines

WT and *Tsc1-/-* MEF cells were obtained from Dr. Kwiatkowski at Harvard University [66]. The following cell lines were purchased from ATCC, Manassas, VA, USA; HEK293 (ATCC CRL-1573), MCF7 (ATCC HTB-22), MDA-MB361 (ATCC HTB-27). WT, *Tsc1-/-* MEFs, HEK293, MCF7, MDA-MB361 cells were cultured in High Glucose (4.5 g/L) Dulbecco’s Modified Eagle Media (DMEM) (Gibco, Waltham, MA, USA), in supplements with 10% FBS (Gibco), 100 ug/mL streptomycin, and 100 U/mL penicillin. All cell lines were cultured at 37 °C with 5% CO_2_. 

### 4.2. Chemicals

Cycloheximide (No. 14126) and Torin 1 (No. 10997) were purchased from Cayman Chemical, Ann Arbor, MI, USA (No. 14126) and reconstituted in DMSO.

### 4.3. Plasmids

cDNAs of mouse *Sirt2* long-form (NM_022432), *Sirt2* short-form (NM_001122765), mouse *mdm2* long-form (NM_010786.4), *mdm2* short-form (NM_001288586.2), mouse *Adnp* long-form (NM_009628.3), and *Adnp* short-form (NM_001310088.1) were cloned into the p3xFlag-PGK plasmid.

### 4.4. Real-Time PCR (RT-PCR) Analysis 

Total RNAs were isolated using the Trizol method recommended by the manufacturer’s protocol. cDNAs were created using reverse transcription using Oligo-d(T) and superscript III (Thermo Fisher Scientific, Waltham, MA, USA). cDNA templates were used to determine alternative splicing isoform expression in a gel-based RT-PCR analysis. Isoform expression levels were determined using densitometry.

### 4.5. List of Primers Used for RT-PCR

Mouse *Ranbp3* RT-PCR forward: 5′-AAGCCTGCCGTCGCACCGTCTGTCT; mouse *Ranbp3* RT-PCR reverse: 5′-CTTCTCCGGCTTCGGGACTGGAGC-3′; mouse *Mdm4* RT-PCR forward: 5′-GCTAAGAAAGAATCTTGTTACATCAGC-3′; mouse *Mdm4* RT-PCR reverse: 5′-ATGTCGTGAGGTAGGCAGTGTGTGA-3′; mouse *Syce2* RT-PCR forward: 5′-AGCATCGGCAGAGTGAGAAC-3′; mouse *Syce2* RT-PCR reverse: 5′-CCGTTTCCACAGTTTGGCAG-3′; mouse *Prrc2b* RT-PCR forward: 5′-GTTGAAAGGCTTCCACTTTGCCGA-3′; mouse *Prrc2b* RT-PCR reverse: 5′-CAGGCACCTTCAGGCTTTGCTTC-3′; mouse *Asph* RT-PCR forward: 5′-CGCAGAACCATCCAAATGACA-3′; mouse *Asph* RT-PCR reverse: 5′-GACCCCTTCATGCTCTGAGG-3′; mouse *Map2k7* RT-PCR forward: 5′-CCGCAGGAGGATCGACCTCAACTT-3′; mouse *Map2k7* RT-PCR reverse: 5′-GGAGCTCTCTGAGGATGGTGAGCG-3′; mouse *Pcdh19* RT-PCR forward: 5′-TACCTGTCCCCAGCTCTTGATG-3′; mouse *Pcdh19* RT-PCR reverse: 5′-AAGGGAGGAGCAACTGACAACAT-3′; mouse *Rpl12* RT-PCR forward: 5′-TCAAGGAGATCCTGGGTACTGC-3′; mouse *Rpl12* RT-PCR reverse: 5′-CCCCAAAAAGCAGCCAAGCC-3′; mouse *Wbp1* RT-PCR forward: 5′-GCCCTGTTCCAACCGGTTCACTGCTTG-3′; mouse *Wbp1* RT-PCR reverse: 5′-GGGGTATGTGAACTGGGCTCAATCCTG-3′; mouse *2310033P09Rik* RT-PCR forward: 5′-GACAGCTGCTCCTCATCTCC-3′; mouse *2310033P09Rik* RT-PCR reverse: 5′-GGTGTCCTTTGGTGTAGCCA-3′; mouse *Anks1* RT-PCR forward: 5′- GCTGACTCGAAAGGCTGCTACC-3′; mouse *Anks1* RT-PCR reverse: 5′- TCCAGGGGCGTTTCAAACTTGTTG-3′; mouse *Ect2* RT-PCR forward: 5′- GAAATGCCGCAGGTTGAAGCAAG-3′; mouse *Ect2* RT-PCR reverse: 5′- ACTGGTGGCCCAACAATCCTACA-3′; mouse *Adgrl2* RT-PCR forward: 5′- TCCTCTGTGAGGCTGATGGAAC-3′; mouse *Adgrl2* RT-PCR reverse 5′- CAACAACATTGTGGCTGTGTGCG-3′.

### 4.6. siRNA Knockdown and Antibodies

Cells were transfected with siRNAs synthesized by Integrated DNA Technologies (IDT) using RNAiMax Transfection reagents (Thermo Fisher Scientific) recommended by the manufacturer’s protocol. siRNA targeting mouse *Srsf3*: 5′-CGUGAUAUCAAGAAUUGU-3′. The knockdown efficiency of Srsf3 was determined by qPCR with forward primer 5′-GCTGCCGTGTAAGAGTGGAA-3′ and reverse primer 5′-AGGACTCCTCCTGCGGTAAT-3′. The antibodies used for Western blot analysis include anti-Flag (No. F3165, Sigma Aldrich, Burlington, MA, USA), anti-Actin (No. 612657 BD Biosciences, Durham, NC, USA), anti-4EBP1 (#9644, Cell Signaling Technology, Danvers, MA, USA), anti-phospho-S6 (#4857, Cell Signaling Technology), anti-S6 (sc-74576, Santa Cruz Biotechnology, Dallas, TX, USA), anti-Tubulin (12004166, Bio-Rad, Hercules, CA, USA), Anti-Secondary antibody against mouse: goat-anti-mouse IgG-HRP (No. sc-2005, Santa Cruz Biotechnology).

### 4.7. Protein Stability

Cells were treated with 30 ug/mL cycloheximide to inhibit protein synthesis and were harvested at different time points. Cells were then lysed and 15–30 ug of lysates (depending on the expression level of the Flag-tagged protein) were run on SDS PAGE gels for Western blot analysis. 

### 4.8. RNA-Seq Data Analyses

MEFs were treated with either DMSO (control) or Torin 1 at 50 nM for 24 h. Human cancer cell lines were treated with either DMSO or 100 nM Torin 1 for 24 h. To evaluate the transcriptome features under mTOR-hyperactivation at single-nucleotide resolution, we performed RNA-Seq analyses of poly(A+) RNAs isolated from WT, *Tsc1-/-* MEFs, MCF7, and MDA-MB361 cells. Paired-end reads were aligned to the mouse mm10 reference genome or the human hg19 reference genome using TopHat2 [67] with up to two mismatches allowed. Kallisto [68] was applied to quantify gene expressions with RefSeq annotation [69].

### 4.9. AS-Quant (Alternative Splicing Quantitation) Pipeline

Alternative splicing analysis was performed using the AS-Quant pipeline [37]. Briefly, the pipeline categorizes the alternative splicing events into the cassette exon, mutually exclusive, alternative 5′ splice site, and alternative 3′ splice site. Reads (*n*) from these alternatively spliced exons were measured and compared to reads (*N*) coverage of the rest of the transcripts. A 2 × 2 Chi-Square test was performed to determine the statistical significance for each alternative splicing event using the ratio of *n*/*N*. Events with a *p*-value < 0.1 and a ratio difference >0.1 were considered real alternative splicing exons.

### 4.10. De Novo Intron Retention Analysis

The input samples were aligned with the reference genome to be converted into BAM files and the read coverage files were generated for each chromosome. For the detection and analysis of both the annotated and novel intron retention events, our pipeline generated a predefined list of independent introns, which do not overlap with any exons on any isoforms of that gene. Using the RefSeq annotation, the independent introns were generated by merging all the annotated exons of a specific gene and subtracting the merged region from the total genomic region. In general, the read coverages in the retained introns are distributed flatly. Based on this speculation, we integrated several measures to characterize the intronic region as the retained one. For each pre-generated independent intron, we considered five different regions (A, B, C, D, and E), which span over the intron and its surrounding exons. A, B, and C are the average read coverage of the 100 bp-long regions, where A and C are from the upstream and downstream boundary regions of the targeted intron, and B is from the middle position of the range, respectively. D and E denote the average read coverage of the upstream and downstream exons surrounding the target intron. To make the computation consistent, intron lengths smaller than 100 were excluded from the analysis. To characterize the ‘flatness’ of the introns and to filter out the false positives, several filtering criteria were introduced in the analysis: (a) the minimum reads spanning over the 3 regions (A, B, and C) >20, (b) the change in the read coverage of the 3 regions (A, B, and C) does not exceed 30% of the highest read coverage, and (c) the average read coverage of the regions D and E is at least 5 times larger than the average read coverage of the sections A, B, and C.

### 4.11. Pfam Domain Analysis

Pfam-Scan was used to search Pfam databases for the matched Pfam domains on each transcript with a 1 × 10^−5^ e-value cutoff [70]. Only the Pfam domains which were overlapped with skipped exon(s) were considered for further analysis. The InterPro Protein Families Database [71] provided the information to link the Pfam domains to the Gene Ontology terms. The LinkDB under the GenomeNet Database [72] provided the information to link the Pfam domains to the KEGG pathways.

### 4.12. Quantification and Statistical Analysis

Signal intensity from gel electrophoresis or Western blots was quantified using the Ver 5.2 of the ImageStudio software (LI-COR, Lincoln, NE, USA). Statistical analysis was performed using the two-tailed Student’s *t* test unless stated otherwise.

## Figures and Tables

**Figure 1 ijms-23-12416-f001:**
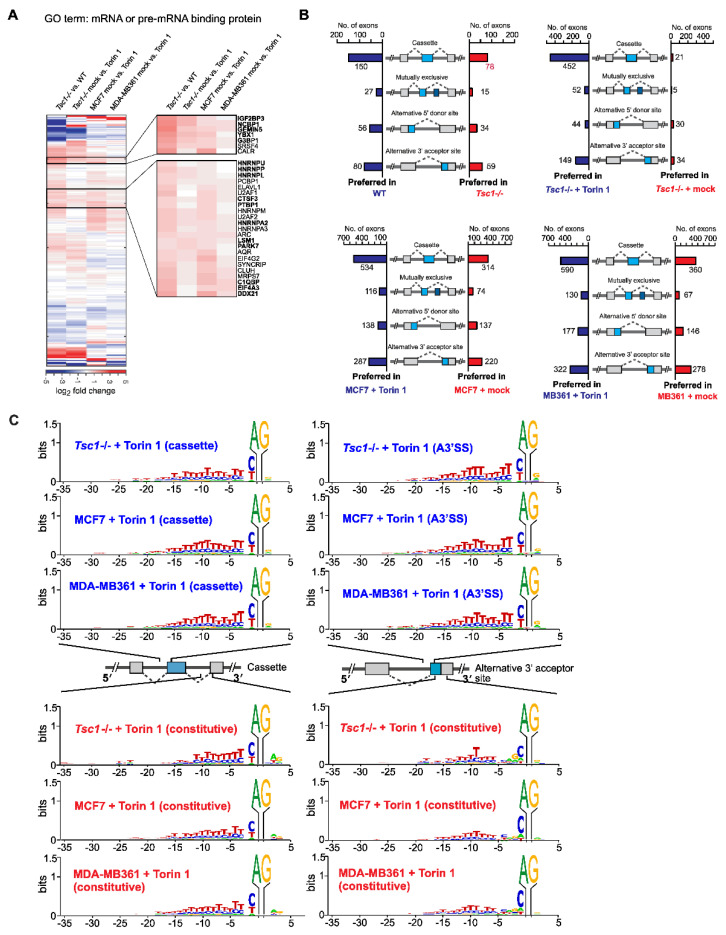
The mTOR-activated transcriptome features widespread exon skipping. (**A**) Hierarchical clustering of select RNA-binding proteins (RBPs) using differential gene expression analysis in low and high mTOR cellular environments. Comparisons were made in the combination of cells as indicated. The x-axis indicates the 285 different RBPs analyzed. The y-axis shows the combination of comparisons. The expression level is color-coded as shown in the scale bar and represents the log2 fold changes of gene expression. (**B**) Analyses of four different types of alternative splicing events in WT vs. *Tsc1-/-* MEFs; *Tsc1-/-* MEFs mock vs. Torin 1 treatment; breast cancer cell line MCF7 mock vs. Torin 1 treatment; MB361 mock vs. Torin 1 treatment. Cases of alternative exon inclusion are compared and illustrated in the combination as indicated in the figure. (**C**) The frequency of sequence preference of the 3′-splice site at the AS exon compared to constitutively spliced exons for *Tsc1-/-*, MCF7, and MDA-MB361 mock- and Torin 1-treated cells. The x-axis represents the nucleotide position relative to the 3′ AG dinucleotide and the y-axis represents the certainty of the nucleotide.

**Figure 2 ijms-23-12416-f002:**
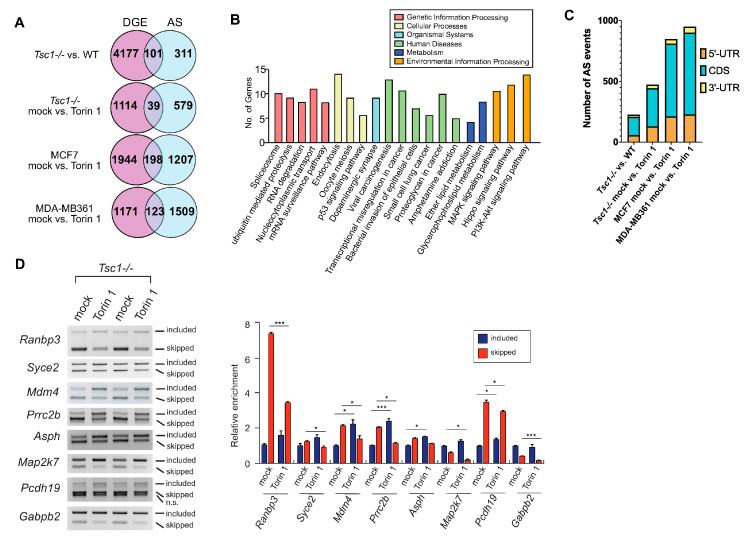
The mTOR-activated transcriptome features widespread exon skipping (cont’d). (**A**) Venn diagram illustrating the unique genes with differential AS and the genes overlapped with mTOR-activated differential expressed genes in the four pairs of cell lines with modulated mTOR activity. (**B**) KEGG pathway enrichment analyses performed on genes found to be differentially spliced by mTOR activation using G:profiler [38]. (**C**) Distribution of mTOR-regulated cassette exon events in different regions of the mRNA. (**D**) RT-PCR and gel electrophoresis of select genes with differential AS changes in *Tsc1-/-* MEFs mock vs. Torin 1 treatment. Semi-quantification of the band intensities was performed using densitometry and statistical significance between mock and treatment samples was determined by Student’s *t*-test. * denotes significant with *p*-value < 0.05; *** for *p*-value < 0.001.

**Figure 3 ijms-23-12416-f003:**
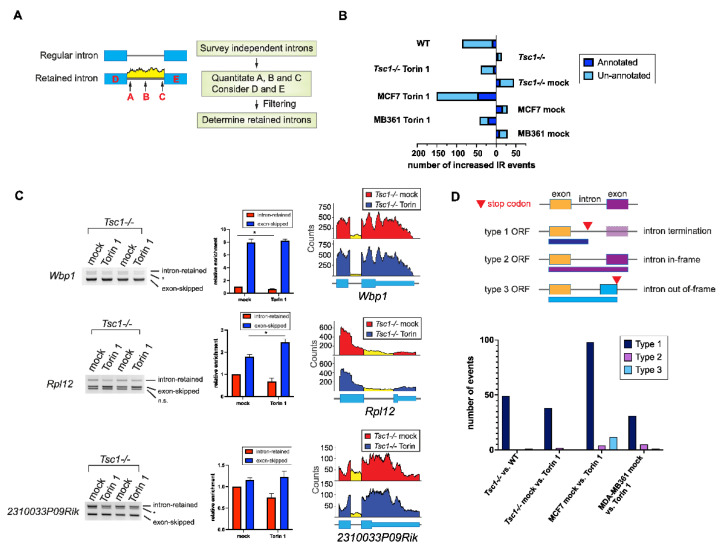
Computational pipeline uncovers novel, unannotated differentially expressed intron-retained transcripts regulated by mTOR signaling. (**A**) A flowchart demonstrating the computational pipeline to identify unannotated retained introns. (**B**) Analyses of the total differentially intron-retained transcripts (both annotated found by AS-Quant and the unannotated) by mTOR signaling in the four pairs of RNA-sequencing datasets. (**C**) RT-PCR gel electrophoresis of select transcripts with differential intron retention found in *Tsc1-/-* MEFs mock vs. Torin 1 treatment. The faint band marked by * could come from another isoform with cryptic splice site usage or another unknown novel transcript. Semi-quantification of the band intensities was performed using densitometry and statistical significance between mock and treatment samples was determined by Student’s *t*-test. * denotes significant with *p*-value < 0.05. (**D**) Count of the different translational outcomes by intron-retention in the four pairs of RNA-sequencing datasets.

**Figure 4 ijms-23-12416-f004:**
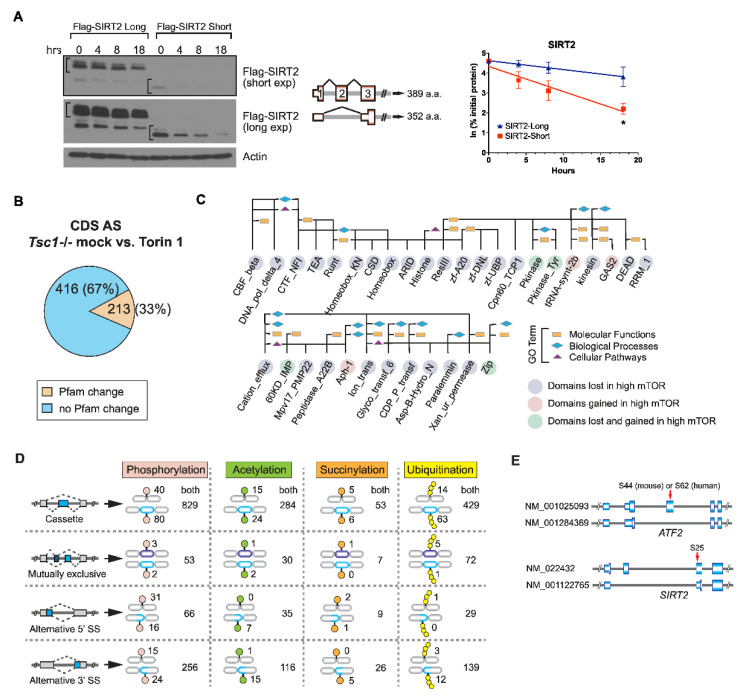
mTOR-driven exon skipping in CDS confers multifarious features in the proteome. (**A**) Analysis of protein isoform stability by Western blotting. Flag-tagged SIRT2 protein isoforms were transiently expressed in HEK293 and the difference in their stabilities was monitored in the presence of cycloheximide (30 ug/mL) for the indicated time points. Actin was used as a loading control. The protein level was quantified using densitometry in ImageStudioLite software and normalized to the actin level. Means (SD) from two technical repeats were subjected to two-tailed Student’s *t*-test for statistical analysis. *p* < 0.05 as significant (*). n.s. denotes no significance. (**B**) Proportion of mTOR-regulated AS from the coding region showed a Pfam domain change. (**C**) Examples of affected Pfam domains and their associated GO terms by mTOR-driven AS events. Bioinformatics analysis of the functional Pfam domains disrupted or gained by AS events in the CDS of mRNA was performed. Pink: Pfam domains gained in hyper-activated mTOR. Light purple: Pfam domains lost in hyper-activated mTOR. Green: Pfam domains affected in both low and high mTOR environments. Rectangle: molecular functions; diamond: biological processes; triangle: cellular pathways. (**D**) Various post-translational modification (PTM) sites found on protein isoforms generated by mTOR-regulated AS. Protein isoforms produced by four types of AS are presented from top to bottom. Four types of PTM (phosphorylation, acetylation, succinylation, and ubiquitination) were interrogated using published mass spectrometry datasets. The number of unique PTM sites found in either exon-included or exon-skipped protein isoforms is summed. The number of common PTM sites located in both protein isoforms is also reported. (**E**) Two examples of published unique functional phosphorylation events found on mTOR-regulated AS protein isoforms of SIRT2 and ARF2.

**Figure 5 ijms-23-12416-f005:**
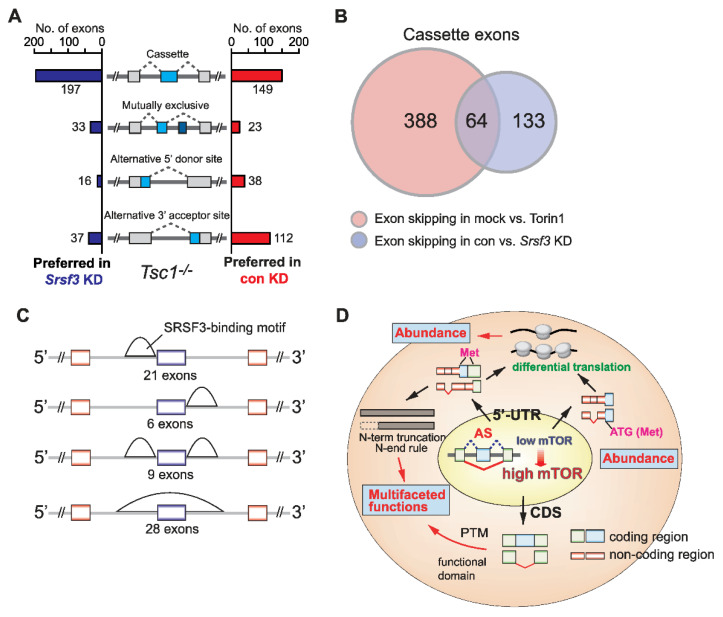
SRSF3 suppresses exon inclusion in mTOR-activated cells. (**A**) Analyses of four different types of alternative splicing events in control vs. *Srsf3* knockdown in *Tsc1-/-* MEFs. Cases of alternative exon inclusion are compared and illustrated. (**B**) Venn diagram illustrating the overlap of skipped exons in mock vs. Torin 1 in *Tsc1-/-* MEFs and control vs. *Srsf3* knockdown in *Tsc1-/-* MEFs datasets. (**C**) In silico prediction of SRSF3 putative binding sites using the overlapped exons as reported in panel B. The putative sites were determined using RBPMap [58]. (**D**) Summary model of mTOR-mediated AS and the impact on the functional proteome.

## Data Availability

RNA-Seq raw data and the description of data can be found in the following link. https://www.ncbi.nlm.nih.gov/sra/?term=SRP056624 (accessed on 27 March 2015) and https://www.ncbi.nlm.nih.gov/bioproject/886626 (accessed on 3 October 2022). The accession numbers for the RNA-seq data are SRP056624 and PRJNA886626.

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
