# Peer review of "mTOR Contributes to the Proteome Diversity through Transcriptome-Wide Alternative Splicing"

_ijms, 2022, doi:10.3390/ijms232012416_

Round 1
Reviewer 1 Report
This is an interesting and well-designed paper from Yong and colleagues. The authors showed that mTOR hyperactivation promotes exon skipping/exclusion, resulting in short isoform production. They further identified that upregulation of splicing factor SRSF3 upon mTOR activation suppresses exon inclusions, presumably by competing with PTBP1, PTBP3, and HNRNPC for polypyrimidine tracts binding. Using their own bioinformatics tool (AS-Quant), differential AS events were classified based on the RNA-seq data from the mTOR activated cells (Tsc1-/- MEFs or Torin 1 treatment). In addition, the computational pipeline identifies novel and unannotated differentially expressed intron-retained transcripts regulated by mTOR signaling. Finally, they demonstrated that the exon skipped mRNAs upon mTOR-activation produced short proteins that lack functional domains, particularly for the regulatory post-translational modifications. The authors have put forth an attractive and novel phenomenon and there are considerable data supporting it. This study provides molecular insights into mTOR-driven transcriptome-wide AS events and proteome regulation. Therefore, I recommend publishing this paper without additional revisions.
Author Response
We thank the reviewer for reading and evaluating this manuscript.
Reviewer 2 Report
The manuscript by Cheng et al. entitled "mTOR contributes to the proteome diversity through transcriptome-wide alternative splicing” characterized the role of mTOR promotes transcriptome-wide exon-skipping/exclusion, producing short isoform transcripts from genes. There are still some suggestions to improve the quality of the manuscript although the findings in this manuscript are potentially interesting.
1. Figure 1A, Why the expression of SRSF3 didn't change while the authors believe that SRSF3 is the most important splicing regulator?
2. Figure 2D,Figure 3C, the authors should use Sanger sequencing to exam the detail sequence of the n.s bands
3. Significance calculation should be included in the figures.
4. Page 6, the detail information of alternative spliced SIRT2 and ARF2 should be shown in the figure.
5. The summary model is too complicated and hard to understand.
Author Response
The manuscript by Cheng et al. entitled "mTOR contributes to the proteome diversity through transcriptome-wide alternative splicing” characterized the role of mTOR promotes transcriptome-wide exon-skipping/exclusion, producing short isoform transcripts from genes.
There are still some suggestions to improve the quality of the manuscript although the findings in this manuscript are potentially interesting.
1. Figure 1A, Why the expression of SRSF3 didn't change while the authors believe that SRSF3 is the most important splicing regulator?
Previously, we showed that 3’-UTR alternative polyadenylation (APA) in SRSF3 increases upon mTOR signaling and promotes the expression of SRSF3 protein without changing its mRNA level (Nucleic Acids Res. 2019 Nov 4;47(19):10373-10387. doi: 10.1093/nar/gkz761). In that study, we further showed that SRSF3 was one of the splicing factors in suppressing exon inclusion in AS. Despite the hundreds of other RBPs found to be differentially expressed by mTOR signaling (shown in Figure 1A), we chose to focus on SRSF3 because of its known primary role in alternative splicing regulation (particularly suppression of exon inclusions) while other RBPs have fewer specific roles in RNA processing. Additionally, SRSF3 has a consensus recognition motif near/at the 3’ splice site, which made us believe its major involvement in mTOR-driven splicing regulation. In this study, while we mainly focused on SRSF3’s role in mTOR-regulated AS, we believe that mTOR-regulated AS is coordinated by many RBPs. We will continue to study other overexpressed RBPs and their roles in mTOR-regulated AS in the future.
2. Figure 2D,Figure 3C, the authors should use Sanger sequencing to exam the detail sequence of the n.s bands
We have sent out the n.s. bands for Sanger sequencing. The sequencing results and blast analyses showed that the n.s. bands from Rpl12 and Pcdh19 did not align with any region of the gene of interest, confirming that the tested band is indeed non-specific. However, the tested band from Wbp1 and 2310033P09Rik that we thought were non-specific actually aligned to Wbp1 and 2310033P09Rik, respectively. Since these bands belong to a shorter version of the intronretained transcript, we think that they are coming from incomplete splicing reaction and/or partial splicing of the retained intron. Nonetheless, we think that due to the overall unchanged level of tested band intensity, it should not interfere with our alternative splicing analysis. Based on these, our conclusions from these experiments stay the same.
3. Significance calculation should be included in the figures. (completed)
Thanks for pointing this out. We now have incorporated statistical analyses into the RT-PCR quantification plots in the revised figure.
4. Page 6, the detail information of alternative spliced SIRT2 and ARF2 should be shown in the figure.
Thanks for this suggestion. We have added a detailed diagram illustrating the unique phosphorylation events on SIRT2 and ARF2 isoforms (Figure 4E). We also found an error in the text about the ARF2 unique phosphorylation site and corrected the information in the text (line 223-225).
5. The summary model is too complicated and hard to understand.
Thanks for the suggestion. We have updated the summary model figure and included more descriptions in the discussion section to make it easier to understand (line 286-291).

Reviewer 3 Report
In this article Cheng et al add a meaningful piece of information about mTOR regulation of cell proteome by using three different cellular models. They found that mTOR regulates expression levels of different RNA binding proteins (RBPs), among which SRSF3 and several alternative splicing events, particularly cassette exons important for the function and abundamce of the correspondent protein product.
While this article is well written and designed, there are some queries that need to be addressed and clarified.
1. Authors must show the level of mTOR protein in the three analyzed cell models and its level of activation (through western blot or IF analysis of downstream targets such as 4E-BP1 and its phosphorylated form/pS6 expression..). They can at least cite their previous paper for mTOR levels in MEFs but having here a new repeated experiment with the mTOR levels in MEFs compared to MCF7 and MDA-MB361 (mock or Torin1 treated) could help to appreciate differences between the two systems (MEFs and breast cancer cells).
2. In paragraph 2.4 authors should show efficient knock down of Srsf3 in Tsc1 -/- MEFs throught qPCR and Western Blot analysis
3. Material and Methods section should be improved
While for cycloheximide treatment details on the concentration and timing are specified in the main text, for Torin1 they are not, so they must be added.
For western blot authors should state the quantity of protein loaded.
4. Authors found protein instability of the shorter isoforms of the three genes analyzed and attribute this result to protein degradation, but is there also a contribution of instability of the shorter transcripts compared to the longer ones?
Minor comments
Lane 68 Use the more common nomenclature “ alternative 3’ ss” and “alternative 5’ ss” instead of “3’ AS” “5’AS”
Lane 104 and 143 “selected” instead of select
Author Response
In this article Cheng et al add a meaningful piece of information about mTOR regulation of cell proteome by using three different cellular models. They found that mTOR regulates expression levels of different RNA binding proteins (RBPs), among which SRSF3 and several alternative splicing events, particularly cassette exons important for the function and abundance of the correspondent protein product.
While this article is well written and designed, there are some queries that need to be addressed and clarified.
1. Authors must show the level of mTOR protein in the three analyzed cell models and its level of activation (through western blot or IF analysis of downstream targets such as 4E-BP1 and its phosphorylated form/pS6 expression). They can at least cite their previous paper for mTOR levels in MEFs but having here a new repeated experiment with the mTOR levels in MEFs compared to MCF7 and MDA-MB361 (mock or Torin1 treated) could help to appreciate differences between the two systems (MEFs and breast cancer cells).
As suggested by the reviewer, changes of cellular mTOR activity can be monitored by western blot analysis on phosho-S6 and phospho-4EBP1. To compare the changes of mTOR activity in the cell lines we used for RNA-Seq experiments, we cultured Tsc1-/- MEFs, MCF7, and MDAMB361 cells in the absence and presence (24 hours with 50nM, 100nM and 100 nM
respectively) of Torin 1. 20 ug of cellular extracts were loaded into each lane. Western blot analysis showed that mTOR activation levels varied across the cellular models tested (Figure S1). Interestingly, 4EBP1 and S6 protein reacted differently to Torin 1 treatment depending on cell types. But overall, we found that Tsc1-/- MEFs and MDA-MB-361 cell lines reacted to Torin 1 more effectively than MCF7 cells. We also found that MCF7 showed a smaller decrease of p-S6 and p-4EBP1 band intensity upon Torin 1 treatment compared to Tsc1-/- MEFs and MDA-MB361 cells, indicating that there seem to be differences in the mTOR activation across cell lines.
2. In paragraph 2.4 authors should show efficient knock down of Srsf3 in Tsc1 -/- MEFs throught qPCR and Western Blot analysis
When we conducted RNA-Seq experiments, we tested the knockdown efficiency of SRSF3 by qPCR. We now show the qPCR result we produced at that time (Figure S3A), which indicates 75% reduction of SRSF3 expression. Unfortunately, we did not conduct the western blot analysis at that time. We do not think that conducting a new RNAi experiment would be meaningful because RNAi efficiency would differ in each experiment. Thus, we apologize that we are only
showing the qPCR result of the same sample but we believe that this would address the reviewer’s concern.
3. Material and Methods section should be improved
While for cycloheximide treatment details on the concentration and timing are specified in the main text, for Torin1 they are not, so they must be added.
Thanks for pointing this out. We have updated the Methods section detailing the treatment concentration and duration for Torin 1 in all of the cell lines (line 419-420).
For western blot authors should state the quantity of protein loaded.
We have updated the Methods section to include the total amount of protein lysate loaded for the western blot analysis (line 414-415).
4. Authors found protein instability of the shorter isoforms of the three genes analyzed and attribute this result to protein degradation, but is there also a contribution of instability of the shorter transcripts compared to the longer ones?
Thanks for this point. For protein stability assays, we ectopically expressed the Flag-tagged cDNA plasmid into HEK293 cells for transient protein expression of the Flag-tagged isoform. As we did not focus on the endogenous AS mRNA/protein isoforms, we believe that the stability of the endogenous transcripts does not play a role in our protein stability assay. If we were concerned with the effect of alternative splicing on the transcript stability, we would definitely need to conduct an Actinomycin D pulse and chase experiment. However, our focus was more on the stability of long and short protein isoforms.
Minor comments
Lane 68 Use the more common nomenclature “ alternative 3’ ss” and “alternative 5’ ss” instead of “3’ AS” “5’AS” (completed)
We have revised the text. Thank you for pointing this out.
Lane 104 and 143 “selected” instead of select (completed)
We have revised the text. Thank you for pointing this out.
Reviewer 4 Report
The authors have interrogated the mTOR activated transcriptome and discovered that the hyperactivation of mTOR promoters alternate splicing. The manuscript is well planned and executed. The figures beautifully depict the results.
Author Response
We thank the reviewer for reading and evaluating this manuscript
Round 2
Reviewer 3 Report
The authors responded to all my inquires, so I recommended that the revised paper will be accepted.